# Photonic KAN: a Kolmogorov-Arnold Network Inspired Efficient Photonic Neuromorphic Architecture

**Yiwei Peng\*, Sean Hooten\*, Thomas Van Vaerenbergh, Xian Xiao, Marco Fiorentino, and Raymond Beausoleil**

Hewlett Packard Labs, 820 N. McCarthy Blvd., Milpitas, CA 95035, USA

\* Equal Contributions

{yiwei.peng, sean.hooten}@hpe.com

## Abstract

Photonic analog accelerators offer a promising shift in AI hardware, potentially improving inference bandwidth, latency, and power consumption by several orders of magnitude over digital counterparts. Recently, Kolmogorov-Arnold Networks (KAN) models were introduced, demonstrating enhanced parameter scaling and interpretability compared to traditional multilayer perceptron (MLP) models. Inspired by the KAN architecture, we propose the Photonic KAN – an integrated all-optical neuromorphic platform leveraging highly parametric nonlinear transfer functions along KAN edges to overcome key limitations in photonic neural networks. In this work, we implement such nonlinearities in the form of cascaded ring-assisted Mach-Zehnder Interferometer (RAMZI) devices. In our test cases, the Photonic KAN showcases enhanced parameter scaling and interpretability compared to existing photonic neural networks. The Photonic KAN achieves approximately 2300× reduction in footprint-energy efficiency, alongside a 7× reduction in latency in function-fitting tasks compared to previous MZI-based photonic accelerators. This breakthrough presents a promising new avenue for expanding the scalability and efficiency of neuromorphic hardware platforms.

## 1 Introduction

Multi-Layer Perceptrons (MLPs) are fully-connected feedforward neural networks, forming the foundation of modern deep learning models. Each node applies an activation function to the weighted sum of its inputs, allowing MLPs to approximate any continuous function with sufficient layers and neurons. Despite their versatility in tasks like classification [1, 2], regression, and natural language processing, MLPs face challenges in interpretability and scaling.

Kolmogorov-Arnold Networks (KANs), introduced in [3], present a compelling alternative to traditional MLPs by learning activation functions on edges. The original KAN model utilizes adaptable B-splines to construct activation functions, replacing linear weight parameters. The advantages of this approach, including scalability, flexibility, efficiency, and interpretability, have spurred further investigation into their potential [4, 5, 6].

Digital electronics hardwares currently dominate the AI accelerator market. Nevertheless, they are encountering substantial obstacles as transistor-based chips struggle to deliver further performance improvements without consuming excessive power. Furthermore, digital electronic computing's clock speed inherently restricts inference bandwidth and has latency bottlenecks [7, 8]. Integrated photonic accelerators performing analog processing are emerging as a powerful alternative, with massive parallelism, sub-nanosecond latency and attojoule multiply-accumulate operation (MAC)

38th Second Workshop on Machine Learning with New Compute Paradigms at NeurIPS 2024(MLNCP 2024).

energy efficiency [9, 10]. Researchers are actively exploring various optical systems, including Mach-Zehnder Interferometers (MZI) meshes [9, 10, 11], microring resonator (MRR) crossbars [12], and coherent MRR networks [13], to implement MLP layers in the photonic domain. Despite their potential, existing photonic accelerators struggle with scalability and implementing traditional ANN architectures. For example, while a single NVIDIA H100 GPU can handle a popular language model GPT-2 XL with 1.5 billion parameters [14], current state-of-the-art optical neural networks are limited to a scale of 64 x 64 or smaller [15].

To overcome these challenges, we introduce Photonic KAN, the first customized photonic accelerator designed for Kolmogorov–Arnold Networks. The main contributions of this work are four-fold:

- **Scalability:** Photonic KAN achieve linear power scaling with network width, thus overcoming the exponential growth limitations of other photonic accelerators.
- **Accuracy:** Photonic KAN outperforms photonic MLP in accuracy and convergence speed, achieving similar performance with just 16% of the parameters on our test case, leading to smaller models and lower computational demands.
- **Efficiency:** Photonic KAN delivers approximately 2× energy efficiency with increased network depth, 75,000× energy efficiency with increased network width, along with 1.35× area and 7× latency reductions compared to prior photonic accelerators with equivalent parameters [9].
- **Interpretability:** Photonic KAN is amenable to pruning, which leads to enhanced interpretability. It allows for easier identification and correction of errors or unexpected behaviours.

To our best knowledge, this is the first framework that supports large-scale demonstrations, over 2300× reduction in footprint-energy than prior art achieving the same performance.

## 2 Preliminaries

The power of KANs lies in their distinctive architecture. In contrast to traditional MLPs, which rely on fixed activation functions at nodes, KANs employ adaptable activation functions on network edges. Based on the Kolmogorov-Arnold Representation Theorem [16], this simplifies learning complex high-dimensional functions to learning a few one-dimensional functions, providing KANs with a flexible and adaptable architecture that adjusts dynamically to intricate data patterns. As a result, a KAN layer with $n_{in}$-dimensional inputs and $n_{out}$-dimensional outputs can be defined as a matrix of 1D functions:

$$\Phi = \{\phi_{q,p}\}, \quad p = 1, \ldots, n_{in}, \quad q = 1, \ldots, n_{out} \tag{1}$$

where the functions $\phi_{q,p}$ are univariate functions having trainable parameters. We denote the $j$-th activation function of $i$-th neuron in the $l$-th layer as $\phi_{l,i,j}$. The activation value of the $(l+1, j)$-th neuron is simply the sum of all incoming activation functions $x_{l+1,j} = \sum_{i=1}^{n_l} \phi_{l,j,i}(x_{l,i})$. Each layer's transformation, $\phi_l$, acts on the input $x_l$ to produce the next layer's input $x_{l+1}$ in matrix form is described as [3]:

$$x_{l+1} = \Phi_l(x_l) = \begin{pmatrix} \phi_{1,1}^{(l)} & \cdots & \phi_{1,n_l}^{(l)} \\ \vdots & \ddots & \vdots \\ \phi_{n_{l+1},1}^{(l)} & \cdots & \phi_{n_{l+1},n_l}^{(l)} \end{pmatrix} x_l. \tag{2}$$

A general KAN network consists of $L$ layers, and hence may be expressed as:

$$\text{KAN}(x) = (\Phi_{L-1} \circ \Phi_{L-2} \circ \cdots \circ \Phi_0)(x). \tag{3}$$

## 3 Proposed Photonic KAN Design

Here we propose a photonic neuromorphic architecture that leverages tunable nonlinear functions along network edges. As shown in Fig. 1(a), the laser light first undergoes input vector modulation, then passes through a 1:N splitter. If the laser power is insufficient, additional lasers can be cascaded to boost it. At the core of our photonic KAN, we introduce an all-optical nonlinear learnable basic block, the MRR-assisted MZI (RAMZI) unit – capable of generating diverse nonlinear functions

through parameter adjustments. We modeled the unit and experimentally validated its functionality. Further measurement details are available in [17].

Nonlinearity originates from the high-Q MRR within the RAMZI structure. Free-carrier dispersion (FCD) accumulates within the MRR, inducing a nonlinear phase shift that the MZI converts into a nonlinear transmission response. When operating at low input power, the MRR's output is minimal near its initial resonant wavelength. However, as input power rises, the resonance undergoes a blueshift, resulting in a brief decrease in output followed by a rapid increase. Crucially, the specific nonlinear function of the MRR can be tailored by adjusting loss and detuning resonance. These parameters can be individually tuned via carrier injection and thermal effects. Metal–oxide–semiconductor capacitor (MOSCAP) phase tuning presents a promising alternative, eliminating static power consumption and boosting efficiency [18]. The RAMZI structure offers enhanced programmability over single MRRs by manipulating the phase of one MZI arm to achieve constructive and destructive interference at varying input powers. This enables efficient realization of diverse nonlinear activation functions (NAFs).

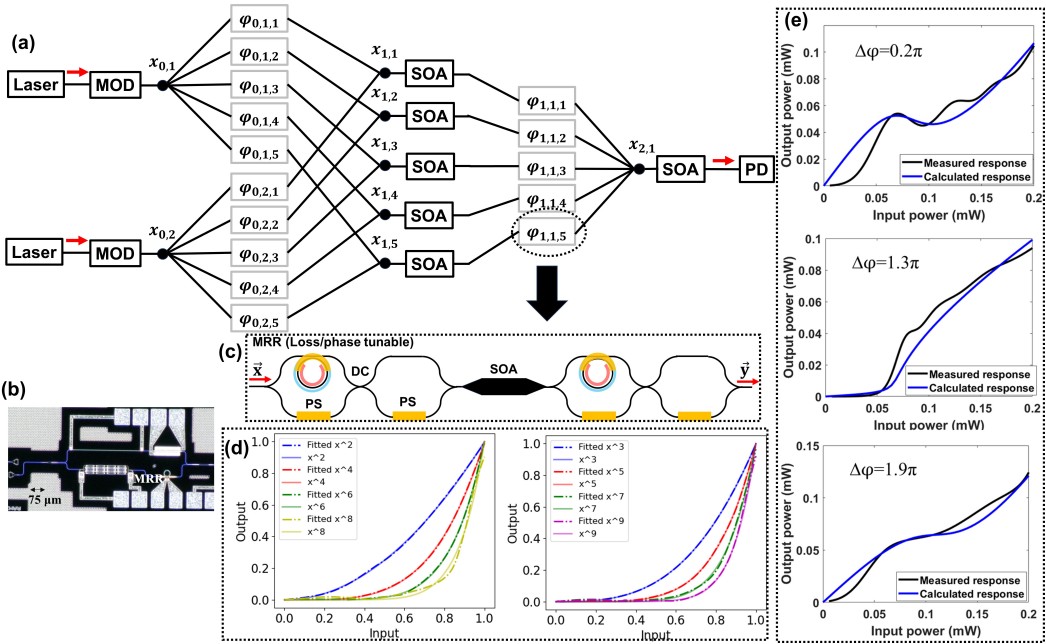

Figure 1: (a) Our proposed photonic KAN. (b) Micrograph of the fabricated RAMZI. (c) Enlarged D-RAMZI unit cell. (d) Expressed polynominal basis with one D-RAMZI. (e) Measured and simulated NAFs for RAMZI at various $\Delta\phi$.

Our experimentally demonstrated RAMZI design, fabricated at Advanced Micro Foundry, confirms the theoretical predictions [17]. As shown in Fig. 1(e), three simulated (blue lines) and measured (black lines) responses are obtained with phase differences $\Delta\phi$. Notably, full programmability is achieved within a low input power range of 0-0.2 mW (-7 dBm), reducing laser power requirements and enhancing network efficiency. However, representing arbitrary NAFs in analog KANs is a fundamental challenge for their effective implementation. In particular, the RAMZI unit is limited to just four parameters: two for adjusting MRR phase and loss, and two for controlling the phase shifter. To expand the range of available nonlinear functions and provide more control over network behavior, we propose a design where two RAMZI units are cascaded within a single unit cell called dual-RAMZI (D-RAMZI), depicted in Fig. 1(c). An semiconductor optical amplifier (SOA) is used between the RAMZI units to guarantee sufficient power for triggering nonlinearity in the second RAMZI. A single D-RAMZI cell demonstrates excellent expressibility, achieving a close fit up to a 9th-order polynomial in Fig. 1(d), making it highly suitable for constructing various nonlinear responses. The photonic KAN accelerator comprises D-RAMZI unit cells acting in parallel as edges between nodes (Fig. 2(a)). This enables fixed loss per layer and minimizes signal travel distance, resulting in significant improvements in power efficiency and latency compared to conventional photonic MZI-based MLP architectures (described in more detail in Sec. 5).

A $N{\times}M$ KAN consists of $NM$ D-RAMZIs with $9{\times}NM + M$ parameters for tuning MRR phase, amplitude, MZI phase, and amplifiers. Correspondingly, the learnable parameters of one MLP layer and one MZI-optical neural networks (ONN) layer are $NM$ and $2NM$, respectively. As we will show, Photonic KANs usually require much smaller network size than Photonic MLPs.

## 4 Results

In this section, we benchmark the ideal performance of the photonic KAN in simulation to characterize its expressivity given the limitations of the analog NAFs. In the next section, we will consider engineering challenges such as loss compensation and scalability. We employed PyTorch to implement the network architecture in our simulations, with architecture depicted in Fig. 1(a). The trainable D-RAMZI NAFs are parameterized using a semi-analytical approach. The MRR nonlinearity was precomputed, with input power swept from 0 to 0.2 mW in 100 steps. To enable differentiation, this nonlinearity was interpolated by sweeping the loss and the phase in 16 steps, respectively. For the remaining D-RAMZI elements, we adopted an analytical approach based on S-matrices, with the phase shifters continuously tunable across a 0 to $2\pi$ range.

### 4.1 MNIST and Fashion-MNIST

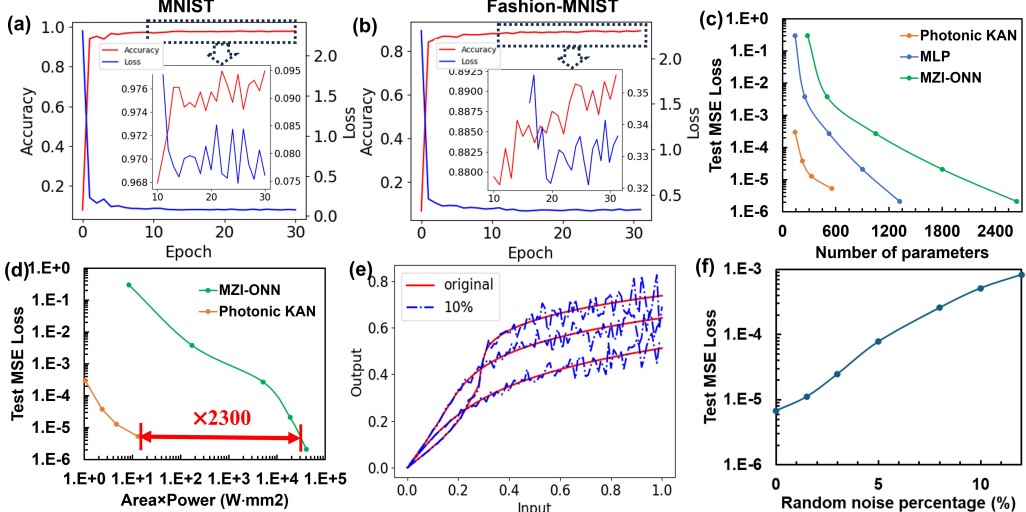

Figure 2: (a) MNIST results. (b) Fashion-MNIST results. (c) Test loss and (d) footprint-energy efficiency for MLP and photonic KAN on the function fitting task. (e) Selected NAFs with 10% random noise. (f) Test loss at different noise levels.

Our simulations utilized a two-layer photonic KAN network with [784, 64, 10] architecture, trained on the MNIST and Fashion-MNIST datasets. We employed Categorical Cross-Entropy loss, the AdamW optimizer with a learning rate of $1 \times 10^{-2}$, and an exponential learning rate scheduler over 30 epochs. As shown in Fig. 2, the network achieved a competitive 98% accuracy on MNIST, and 89% on the more demanding Fashion-MNIST dataset, which are both comparable to conventional KANs [6, 19]. Our photonic architectures demonstrate rapid convergence, achieving over 80% accuracy after a single training epoch. These results highlight the potential of photonic KANs in basic image classification tasks.

### 4.2 Function Fitting

As shown in [20], MLPs perform poorly on high-frequency components, which are crucial for multi-scale partial differential equations (PDEs), image and audio compression, and medical applications. We compare the performance of our proposed photonic KANs against both ideal MLPs and conventional photonic MZI-based ONNs on function-fitting tasks involving high-frequency components. We fitted the function $y = \sin\left(\frac{\pi}{2}x\right) + 0.1\sin(10\pi x)$ using photonic KAN and MLP. As shown in Fig. 2(c), we increase KAN complexity from [1,3,3,1] to [1,5,5,5,1] and MLP complexity from

[1,10,10,1] to [1,20,20,20,20,1], while maintaining the same dataset and optimizer (AdamW). As the results show, the photonic KAN demonstrates faster convergence and higher accuracy with the same number of parameters. It requires around 16% the parameters to achieve comparable performance. In contrast, the unitary properties of MZI architectures complicate building MLPs, requiring MZI-based ONNs to use twice the tunable parameters to match MLP performance.

For the comparison in Fig. 2(d), we calculate the power consumption and area for photonic KAN and photonic Clements's MZI ONN [21]. The photonic KAN improves the footprint-energy efficiency by around 2300× achieving a similar accuracy due to the following reasons: 1. Reduced parameter requirements for the task. 2. Fewer MZIs, decreasing area and power consumption. 3. Shorter paths that exponential lower required laser power. We expand on these results further in Sec. 5.

The noise resistance of the model is evaluated by randomly adding noise to the ideal calculated nonlinear response. Figure 2(e) illustrates three nonlinear responses under varying noise levels, where the added noise is proportional to the magnitude of the response. It can be observed that photonic KANs perform well, achieving an MSE loss of $10^{-2}$ under 5% noise. Further increases in noise introduce overlap between different nonlinear responses, which degrades the model's performance.

## 4.3 Photonic KANs are Prunable

In practice, we often lack a priori knowledge of the underlying data distribution, making it difficult to predefine an ideal network structure. Approaches to determine this shape automatically are desirable. Similar to the original implementation [3], we start from an overparameterized Photonic KAN and leverage sparsity regularization during training followed by pruning. This approach produces significantly more interpretable KANs compared to those without pruning and decreases hardware energy consumption.

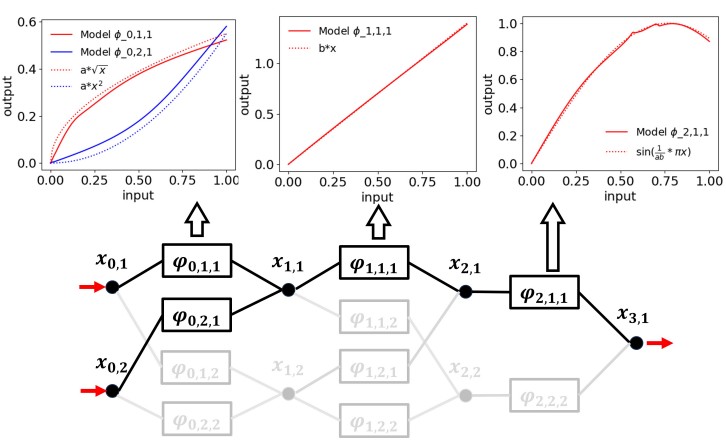

Figure 3: Pruned KAN interprets equations.

Let us consider the simple function fitting task $f(x, y) = \sin\left(\pi\left(\sqrt{x} + y^2\right)\right)$. An ideal [2, 1, 1] KAN is able to express this function perfectly. We train an overparameterized Photonic KAN with sparsification regularization, including L1 regularization and entropy regularization [3]. We augment our regularization strategy by incorporating an additional coefficient, $A(\Phi_l)$, specifically targeting certain parameters, like the amplifier gain. This aims to drive these parameters towards zero, promoting sparsity and simplifying the model. We begin with a fully-connected [2, 2, 2, 1] KAN, uniformly sample 100 points in [0,1] and apply sparsification regularization during training to encourage the network to learn a sparse representation. Subsequent pruning, based on the score thresholds [3], removes 'useless' nodes with weak incoming or outgoing connections. Visualizing the pruned network (Fig. 3) reveals that functions with low magnitudes are effectively faded out, highlighting the important functional components. While our current method cannot remove entire layers due to stability issues, automatic pruning successfully simplifies the KAN to a [2, 1, 1, 1] structure. Importantly, the remaining activation functions visually resemble known symbolic functions ($\sqrt{x}$, $x^2$, $\sin x$) as shown in Fig. 3, making it possible to correctly interpret the mathematical relationships captured by the model. In the hardware itself, 'useless' nodes are deactivated by physically disconnecting them, which is setting their corresponding amplifier gains to zero for our specific architecture.

# 5 Analysis of Photonic KAN Power, Efficiency, Footprint, and Latency

We evaluate the latency, power, area, and energy efficiency of our photonic KAN. Power consumption in static operation comes from six key sources: laser wall-plug power, SOAs, D-RAMZI mesh, modulator and receiver circuits, and analog-to-digital converters (ADCs)/digital-to-analog converters (DACs). SOAs compensate for losses, alleviate the laser's power burden and trigger RAMZI nonlinearity. While on-chip SOAs are the primary power consumers, their operation in the low-gain and linear region minimizes power drive requirements. RAMZI mesh power includes contributions from MZI tuning, MRR tuning, and carrier injection [17]. Transceiver power is based on our 25 GS experiment, covering capacitance charging, drivers, transimpedance amplifiers (TIAs), power amplifiers, and clock power [22, 23]. Our KAN architecture performs analog computation throughout without OEO conversion, needing only high-speed DACs and ADCs for signal modulation and detection. The power consumed by these DACs and ADCs [24, 25], relative to the entire system, is lower than in conventional single-layer photonic accelerators. We determined passive device loss and footprint based on the process design kit (PDK) of a commercial foundry [26].

We compared our KAN with the conventional Clements's MZI ONN [21] and the coherent MZI-Xbar ONN [27] in Fig. 4. On a silicon platform, our photonic KAN reduced power consumption by 35% compared to Clements's ONN and by 50% compared to the coherent MZI-Xbar as network depth increased. These savings are due to fewer MZI devices and reduced path loss. Additionally, we also calculated the power consumption of MZI-based ONNs using all-optical nonlinearity and SOAs. They showed higher power consumption than OEO conversion, as the NAF triggering power is significantly higher than the PD sensitivity, confirming KAN's efficiency stems from its architecture, not the all-optical approach. Additionally, thanks to the shorter optical path, our design is well-suited to the MOSCAP platform, which provides zero static power consumption of MZI and MRR despite high loss. By adopting the MOSCAP platform, our photonic KAN only require half of Clements's ONN power and one-third of the coherent MZI-Xbar power.

The architectural differences between photonic KANs, coherent MZI-Xbar ONNs, and conventional MZI ONNs result in distinct scaling behaviors. Conventional $N \times N$ MZI ONNs require $(2N + 1)$ MZIs, leading to exponential increases in path loss and power consumption as network width grows. In contrast, photonic KANs use a fixed optical path through one D-RAMZI unit per connection, ensuring linear power scaling. Coherent MZI-Xbar ONNs, on the other hand, suffer more from additional losses due to couplers and crossings that grow with network width. As shown in Fig. 4(b), KANs consume 15× less power than coherent MZI-Xbar ONNs and 75,000× less than Clements's MZI ONNs. Additionally, KANs achieve similar accuracy with shallower network depth compared to MZI-based ONNs.

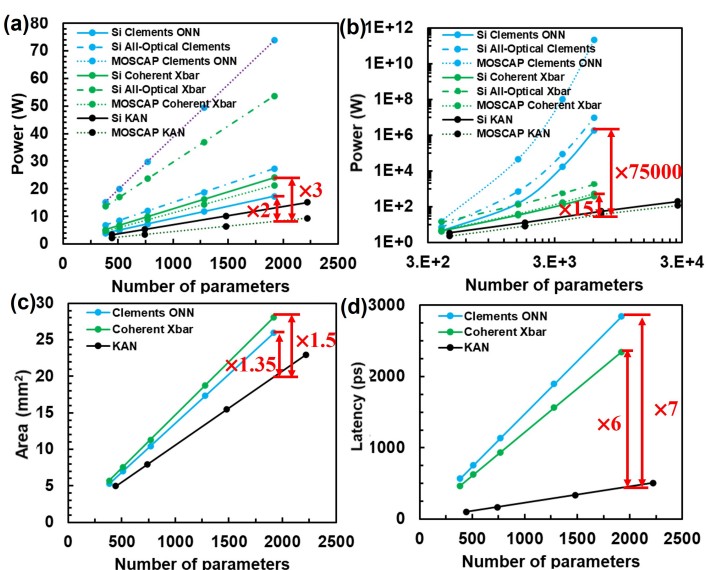

Figure 4: Comparison of (a) power consumption (increasing depth); (b) power consumption (increasing width); (c) footprint (increasing depth); (d) latency (increasing depth).

In terms of footprint, photonic KANs offer a distinct advantage over MZI-based ONNs. This is due to the higher parameter density of RAMZI units. Each RAMZI unit, while occupying a similar footprint to a conventional MZI unit, contains twice the number of tunable parameters (four versus two). As shown in Fig. 4, a photonic KAN matches the parameter count while reducing 35% and

50% space compared to Clements's MZI ONNs and MZI-Xbar ONNs, respectively. Moreover, as shown in Fig. 2(c), photonic KANs achieve similar accuracy with only 16% of the parameters, resulting in a smaller network requirement. Additionally, our KANs are pruning-friendly, allowing further footprint reduction without compromising accuracy.

Reducing latency in ONNs is crucial for unlocking their full potential in real-world applications. The optics latency increases approximately linearly with the size as the optical path increases. The EO/OE latency remains almost the same for each layer. Our photonic KAN design achieves a remarkable 7× reduction in overall latency by eliminating the need for multiple EO/OE conversions and enabling short optical path.

## 6    Conclusion

We introduced the first KAN-inspired photonic neural network, demonstrating its effectiveness in function fitting and image classification. This design overcomes key limitations of existing photonic networks by offering improved accuracy, faster convergence, better scalability and enhanced model interpretability. Photonic KANs requires around 16% the parameters, achieving 2300× footprint-energy efficiency, and 7× lower latency than traditional MZI-based accelerators in function-fitting tasks. While photonic KANs have not yet surpassed electronic accelerators in large-scale demonstrations, overcoming the current challenges in training speed and functional flexibility will unlock their full potential for a wide range of scientific and practical applications.

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
