# OpenReview forum: "Photonic KAN: a Kolmogorov-Arnold Network Inspired Efficient Photonic Neuromorphic Architecture"
_NeurIPS.cc/2024/Workshop/MLNCP — MLNCP Oral_

### Official Review · Reviewer_D6cK · 2024-10-03
**Photonic KAN: a Kolmogorov-Arnold Network Inspired Efficient Photonic Neuromorphic Architecture**

**Rating:** 8
**Confidence:** 4

**Review:**

The manuscript “Photonic KAN: a Kolmogorov-Arnold Network Inspired Efficient Photonic Neuromorphic Architecture” introduces an integrated photonic architecture to implement KANs, which learn activation functions on edges. The non-linear activation functions are implemented using cascaded ring-assisted Mach-Zehnder interferometers. The paper is well-written, and I believe its novel approach will be of great interest to the optical computing research community.​

Comments:

- As the authors note, the architecture cannot represent arbitrary NAFs, as the response of the RAMZI unit is limited to a fixed number of parameters to adjust the phase and loss. It would be interesting to see further theoretical work on the expressibility of the system given these constraints.

- The claim “Photonic KANs requires around 16% the parameters, achieving 2300× footprint- 267  energy efficiency, and 7× lower latency than traditional MZI-based accelerators” in the abstract (line 13-14) and conclusion (266-267) is only shown for a specific application. Since the authors do not provide proof that this is the general case, I suggest revising these sentences to clarify the context.

- How does noise impact the system’s performance, and how does it scale as the system size increases? For instance, if the measured response shown in Fig. 1 is incorporated into the trained simulated system, what effect does it have on the overall performance?

- In Fig 4, is the number of parameters for KAN vs ONN proportional to the performance? For example, do you need fewer parameters in ONNs to achieve the same performance as KANs for certain tasks?

Minor comments:
- In Eq. 2, it may simplify notation to use superscript for the layer index: e.g., $\phi^{(l)}_{m, n}$

---

### Decision · Program_Chairs · 2024-10-10

Accept (Oral)